# PROPER MEASURE FOR ADVERSARIAL ROBUSTNESS

## ABSTRACT

This paper analyzes the problems of adversarial accuracy and adversarial training. We argue that standard adversarial accuracy fails to properly measure the robustness of classifiers. Its definition has a tradeoff with standard accuracy even when we neglect generalization. In order to handle the problems of the standard adversarial accuracy, we introduce a new measure for the robustness of classifiers called genuine adversarial accuracy. It can measure the adversarial robustness of classifiers without trading off accuracy on clean data and accuracy on the adversarially perturbed samples. In addition, it does not favor a model with invariance-based adversarial examples, samples whose predicted classes are unchanged even if the perceptual classes are changed. We prove that a single nearest neighbor (1-NN) classifier is the most robust classifier according to genuine adversarial accuracy for given data and a norm-based distance metric when the class for each data point is unique. Based on this result, we suggest that using poor distance metrics might be one factor for the tradeoff between test accuracy and $l_p$ norm-based test adversarial robustness.

## 1 INTRODUCTION

Even though deep learning models have shown promising performances in image classification tasks (Krizhevsky et al., 2012), most deep learning classifiers are vulnerable to adversarial attackers. By applying a carefully crafted, but imperceptible perturbation to input images, so-called *adversarial examples* can be constructed that cause the classifier to misclassify the perturbed inputs (Szegedy et al., 2013). These vulnerabilities have been shown to be exploitable even when printed adversarial images were read through a camera (Kurakin et al., 2016). Adversarial examples for a specific classifier can be transferable to other models (Goodfellow et al., 2014). The transferability of adversarial examples (Papernot et al., 2017) enables attackers to exploit vulnerabilities even with limited access to the target classifier.

**Problem setting.** *In a nonempty clean input set $\mathcal{X} \subset \mathbb{R}^d$, let every sample $x$ exclusively belong to one of the classes $\mathcal{Y}$, and their classes will be denoted as $c_x$. A classifier $f$ assigns a class label from $\mathcal{Y}$ for each sample $x \in \mathbb{R}^d$. Assume $f$ is parameterized by $\theta$ and $L(\theta, x, y)$ is the cross entropy loss of the classifier provided the input $x$ and the label $y \in \mathcal{Y}$.*

Note that this exclusive class assumption is introduced to simplify the analysis. Otherwise, the definition of adversarial examples (Biggio et al., 2013) may not match with our intuition as explained in Section 1.1.

### 1.1 ADVERSARIAL EXAMPLES

**Definition 1** (**Adversarial Example**). *Given a clean sample $x \in \mathcal{X}$ and a maximum permutation norm (threshold) $\epsilon$, a perturbed sample $x'$ is an adversarial example if $\| x - x' \| \leq \epsilon$ and $f(x') \neq c_x$ (Biggio et al., 2013).*

When exclusive class assumption in the problem setting is violated, different oracle classifiers may assign different classes for the same clean samples. (Oracle classifiers refer to classifiers that are robust against adversarial examples (Biggio et al., 2013) for appropriately large $\epsilon$. Human classifications are usually considered as oracle classifiers.) For example, while many people assign class 7 for the top right sample shown in Figure 1, some people can assign class 1 or 9 because of the ambiguity of that example. If we label data with the most popularly assigned classes, according



Figure 1: Examples of confusing near image pairs with different classes of MNIST training dataset (LeCun et al., 2010). The $l_2$ norms of the pairs are 2.399, 3.100 and 3.131 from left to right. From these examples, we can say the exclusive class assumption in the problem setting can be violated.

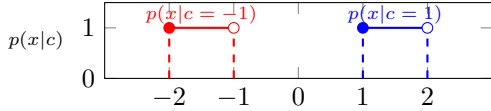

Figure 2: Plots of $p(x|c = -1)$: red and $p(x|c = 1)$: blue for first toy example.

to the definition of adversarial example, even some clean samples will be considered as adversarial examples without perturbing the clean samples according to the classifications of some people.

## 1.2 STANDARD ADVERSARIAL ACCURACY

The following measure is commonly used for comparing different classifiers on vulnerability to adversarial attacks (Biggio et al., 2013; Madry et al., 2017; Tsipras et al., 2018; Zhang et al., 2019).

**Definition 2 (Standard adversarial accuracy).** $\mathbb{1}()$ *is an indicator function that has value* 1 *if the condition in the bracket holds and value* 0 *if the condition in the bracket does not hold. Then, standard adversarial accuracy (by maximum perturbation norm)* $a_{std;\,max}(\epsilon)$ *is defined as follows.*

- $a_{std;\,max}(\epsilon) = \mathbb{E}_{x \in \mathcal{X}} \left[ \mathbb{1} \left( f(x^*) = c_x \right) \right]$ *where* $x^* = \underset{x':\,\|x'-x\|\leq\epsilon}{\arg\max} \ L(\theta, x', c_x).$

### 1.2.1 ONE-DIMENSIONAL TOY EXAMPLE

Even though standard adversarial accuracy by maximum perturbation norm is commonly used as a measure of adversarial robustness (Biggio et al., 2013; Madry et al., 2017; Tsipras et al., 2018; Zhang et al., 2019), it is not clear whether this measure can be used to choose better models. To show this measure is not an appropriate measure of robustness, we introduce the following example.

Let us consider a toy example (see Figure 2) with predefined (pre-known) classes to clean samples in order to simplify the analysis. There are only two classes $-1$ and $1$, i.e., $\mathcal{Y} = \{-1, 1\}$, and one-dimensional clean input set $\mathcal{X} = [-2, -1) \cup [1, 2) \subseteq \mathbb{R}$. $c_x = -1$ when $x \in [-2, -1)$ and $c_x = 1$ when $x \in [1, 2)$. $p(c = -1) = p(c = 1) = \frac{1}{2}$, i.e., we assume uniform prior probability.

Let us define three classifiers $f_1$, $f_2$ and $f_3$ for this toy example (see Figure 3). When step function $\text{step}(x)$ is defined as $\text{step}(x) = \begin{cases} 1, & \text{if } x \geq 0, \\ -1, & \text{if } x < 0 \end{cases}$, let $f_1(x) = \text{step}(x - 1)$, $f_2(x) = 1 - \text{step}(x + 4) + \text{step}(x)$, and $f_3(x) = \text{step}(x)$.

Notice that the accuracy for all three classifiers is 1. However, $f_1$ will not be robust against adversarial attacks as points in $[1, 1+\epsilon)$ can be perturbed to change their classification result. $f_2$ is overly invariant when $x < -4$. The oracle classifier will be $f_3$.

When the change of standard adversarial accuracies by maximum perturbation norm $\epsilon$ were considered (see Figure 4) , $f_1$ shows decreasing standard adversarial accuracy even when $\epsilon < 1$. However,

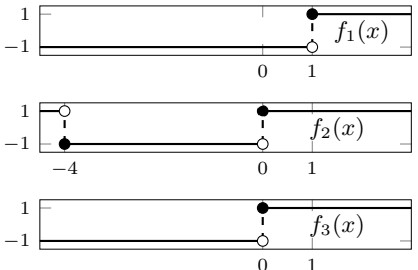

Figure 3: Plots of three classifiers. Top: $f_1(x) = \text{step}(x-1)$. Middle: $f_2(x) = 1 - \text{step}(x+4) + \text{step}(x)$. Bottom: $f_3(x) = \text{step}(x)$ where $\text{step}(x) = 1$ for $x \geq 0$ and $\text{step}(x) = -1$ for $x < 0$.

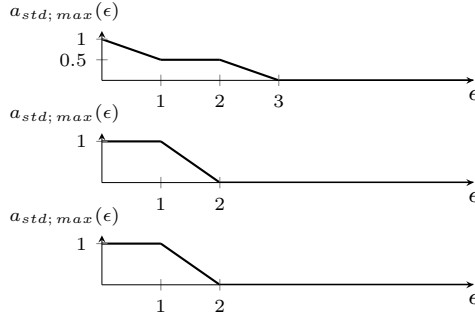

Figure 4: Change of standard adversarial accuracy for $f_1(x)$, $f_2(x)$ and $f_3(x)$ from top to bottom. Observed behaviors of $f_2$ and $f_3$ will be the same when we compare the adversarial accuracy by maximum perturbation norm $\epsilon$.

$f_2$ and $f_3$ are equally robust according to this measure. Thus, the maximum norm-based standard adversarial accuracy function can not tell which classifier is better. (In addition, $f_1$, $f_2$, and $f_3$ have the same Accuracy-Robustness Area (ARA) (Woods et al., 2019), which measures the area under the curve for standard adversarial accuracy plot, that is 1.5. This example shows that ARA is also not a reliable measure for the robustness of classifiers.)

## 1.3 ADVERSARIAL TRAINING

Adversarial training (Goodfellow et al., 2014) was developed to avoid the adversarial vulnerability of a classifier. It tries to reduce the weighted summation of standard loss (empirical risk) $\mathbb{E}\left[L(\theta, x, y)\right]$ and adversarial loss $\mathbb{E}\left[L(\theta, x', y)\right]$, i.e., $\alpha\mathbb{E}\left[L(\theta, x, y)\right] + (1-\alpha)\mathbb{E}\left[L(\theta, x', y)\right]$ where $\alpha$ is a hyperparameter for adversarial training, and $x'$ is an adversarially perturbed sample from $x$ with $\|x' - x\| \leq \epsilon$. By considering both standard and adversarially perturbed samples, adversarial training try to increase accuracies on both clean and adversarially perturbed samples. In the literature on adversarial training, inner maximization of a classifier refers to generating adversarial attacks, i.e., generating adversarially perturbed samples $x^*$ that maximally increases the loss. And outer minimization refers to minimizing the adversarial loss of the model. (Madry et al., 2017) explained that inner maximization and outer minimization of the loss can train models that are robust against adversarial attacks.

However, there is a tradeoff between accuracy on clean data and adversarially perturbed samples even when we neglect generalization of classifiers (Tsipras et al., 2018; Dohmatob, 2018; Zhang et al., 2019). Hence, when we used adversarial training (Goodfellow et al., 2014), we can get a classifier whose accuracy is lower than using non-adversarial training method (Tsipras et al., 2018; Santurkar et al., 2019). Also, Jacobson et al. (Jacobsen et al., 2019) studied samples whose perceptual classes are changed due to perturbation, but not in the model's prediction, what they called "invariance-based adversarial examples." (These examples are also called "type I errors" (Tang et al., 2018) or "invalid adversarial examples" (Stutz et al., 2019).) They found that classifiers trained with adversarial training can be more susceptible to invariance-based adversarial examples. Recently,

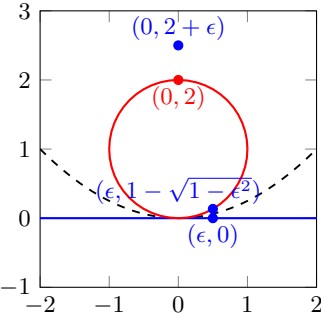

Figure 5: Two-dimensional sunset dataset class A: red and class B: blue. The colored figure shows that the points on class A resemble the shape of the sun and the points on class B resemble the shape of the horizon.

When we consider $l_2$ norm robustness, the decision boundary of the oracle classifier for this example will be a parabola which satisfies $x_2 = \frac{1}{4}x_1^2$ where $x_1$ and $x_2$ represent the first and second axes. That is shown as a dashed black curve. Every point in this parabola has the same distance to the nearest sample in class A and B. This oracle classifier can be trained on two-dimensional data using non-adversarial training, but in higher dimensional sunset datasets, that might not be possible due to high co-dimension (Khoury & Hadfield-Menell, 2019).

further research showed that there is a fundamental tradeoff between adversarial examples (Biggio et al., 2013) and invariance-based adversarial examples (Tramèr et al., 2020).

### 1.3.1 SUNSET DATASET

We introduce a binary classification problem called sunset dataset. This artificial dataset shows the limitation of standard adversarial accuracy and standard adversarial training method (Goodfellow et al., 2014) that is intended to maximize standard adversarial accuracy.

This artificial dataset contains uncountably many points. One class (class A) of the dataset is consists of points on a hypersphere. The other class (class B) is consist of points on a hyperplane contacting with the hypersphere. One important character of this dataset is that infimum of the nearest distance between the two classes is 0.

A two-dimensional sunset dataset is visualized in Figure 5. In that dataset, we can think of a classifier that classifies points inside the circle classified as class A and class B otherwise. The accuracy of the classifier is 1. However, if the maximum perturbation norm $\epsilon > 0$, the classifier will always have adversarial examples (Biggio et al., 2013). For example, point $(0, 2+\epsilon)$ is an adversarial example as it is classified as class B according to the classifier, and $(0, 2)$ has class A. If we try to use adversarial training (Goodfellow et al., 2014), we encounter the problem where point $(\epsilon, 1 - \sqrt{1 - \epsilon^2})$ (or (1,1) when $\epsilon \geq 1$), which is on the red circle, can also be considered as an adversarial example originated from $(\epsilon, 0)$. (We are considering $l_2$ norm robustness. Visualized points are based on $\epsilon = 0.5$) If we change the classes of these points to remove adversarial examples, we simultaneously introduce invariance-based adversarial examples (Jacobsen et al., 2019) on the classifier. If one tries to choose robust classifiers based on standard adversarial accuracy with $\epsilon > 0$, one would choose classifiers with invariance-based adversarial examples over the oracle classifier (mentioned in Figure 5). In other words, standard adversarial accuracy can favor classifiers with more invariance-based adversarial examples.

## 2 GENUINE ADVERSARIAL ACCURACY

**Definition 3** (**Genuine adversarial accuracy**). *We define genuine adversarial accuracy that uses exact perturbation norm. Note that $\mathbb{1}()$ is an indicator function that has value 1 if the condition in the bracket holds and value 0 if the condition in the bracket does not hold. The complement set of a set $S$ will be denoted as $S^c$. $\bar{\mathcal{X}}$ is the topological closure of $\mathcal{X}$, i.e., union of $\mathcal{X}$ and its boundary. Voronoi boundary $VB(\mathcal{X})$ is defined as*

$\left\{x' \in \mathbb{R}^d | \exists x_1, x_2 \in \bar{\mathcal{X}} : x_1 \neq x_2, \|x' - x_1\| = \|x' - x_2\|\right\}$. *Previously allowed perturbation region* $\mathcal{X}_\epsilon$ *is defined as* $\left\{x' \in VB(\mathcal{X})^{\mathsf{c}} | \|x' - x\| < \epsilon \text{ where } x \in \bar{\mathcal{X}}\right\}$.
*Then, genuine adversarial accuracy (by exact perturbation norm)* $a_{gen;\,exact}(\epsilon)$ *is defined as follows.*

- $a_{gen;\,exact}(\epsilon) = \mathbb{E}_{x \in S_{exact}(\epsilon)}\left[\mathbb{1}\left(f(x^*) = c_x\right)\right]$
  *where* $S_{exact}(\epsilon) = \left\{x \in \bar{\mathcal{X}} | \exists x' \in \mathcal{X}_\epsilon^{\mathsf{c}} \cap VB(\mathcal{X})^{\mathsf{c}} : \|x' - x\| = \epsilon\right\}$
  *and* $x^* = \underset{x' \in \mathcal{X}_\epsilon^{\mathsf{c}} \cap VB(\mathcal{X})^{\mathsf{c}} :\, \|x' - x\| = \epsilon}{\arg\max} L(\theta, x', c_x)$ *when* $\epsilon > 0$.

- $a_{gen;\,exact}(0) = \mathbb{E}_{x \in \mathcal{X}}\left[\mathbb{1}\left(f(x) = c_x\right)\right]$ *when* $\epsilon = 0$.

The reason for proposing genuine adversarial accuracy is to avoid using a point $x' \in \mathbb{R}^d$ more than once when calculating a classifier's robustness. That is shown in Theorem 1. $\bar{\mathcal{X}}$ is used instead of $\mathcal{X}$ because using $\mathcal{X}$ can ignore many points in the calculation of robustness (see an example in Section A of the appendix). Voronoi boundary $VB(\mathcal{X})$ is used in the calculation as points on this set can be approached from two different points in $\bar{\mathcal{X}}$ with the same distance, and these points need to be ignored in order to avoid conflicts when measuring adversarial accuracy. Previously allowed perturbation region $\mathcal{X}_\epsilon$ is used to ignore this region for given $\epsilon$. In the calculation of the expected value, $S_{exact}(\epsilon)$ is used because there may be no $x'$ that satisfies new constrain.

Figure 6 shows the result of using genuine adversarial accuracy on the one-dimensional toy example in Section 1.2.1. According to this measure, $f_3$ is the most robust classifier for given data, and that is consistent with the oracle classifier.

**Theorem 1.** *Genuine adversarial accuracy only use each point only once to calculate adversarial accuracy. In other words, it does not allow overlaps.*

The proof is in Section C of the appendix. Theorem 1 indicates that genuine adversarial accuracy avoids the tradeoff between accuracy and (genuine) adversarial accuracy. Also, it no longer favors a classifier with excessive invariance, which indicates that it does not favor a classifier with invariance-based adversarial examples (Jacobsen et al., 2019). The fact that it does not allow overlaps also matches with our intuitive understanding of the robust classification.

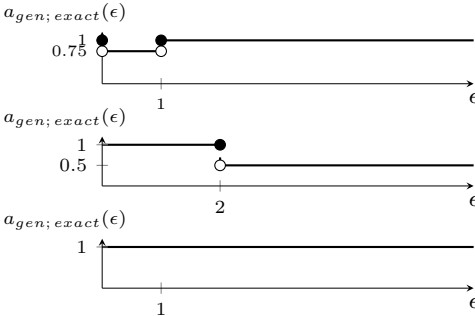

Figure 6: Change of genuine adversarial accuracy $a_{gen;\,exact}(\epsilon)$ for $f_1(x)$, $f_2(x)$, and $f_3(x)$ by exact perturbation norm $\epsilon$ from top to bottom. Details for calculation is explained in Section A of the appendix.

## 3 A single nearest neighbor (1-NN) classifier is the classifier that maximizes genuine adversarial accuracy

**Theorem 2.** *A single nearest neighbor (1-NN) classifier maximizes genuine adversarial accuracy, and it is the almost everywhere unique[1] classifier that satisfies this except for $\bar{\mathcal{X}} - \mathcal{X}$ and Voronoi boundary $VB(\mathcal{X})$.*

Proof of the theorem is in Section D of the appendix. Note that both Theorem 1 and Theorem 2 only hold for the same data. Thus, Theorem 2 does not imply that a 1-NN classifier based on training data will be optimally robust on the test data.

## 4 Possible factor for the tradeoff between test accuracy and $l_p$ norm-based test adversarial robustness: poor distance metric for measuring adversarial robustness

As Theorem 2 only holds for the same data set, it does not directly tell about generalization. Nevertheless, it follows from the theorem that the 1-NN classifier on training data will be the optimally robust classifier on training data as measured by genuine adversarial accuracy. This result gives us an insight that properly applied adversarial training[2] will train models to mimic 1-NN classifiers as their training loss try to enforce them to make the same prediction as 1-NN classifiers. The fact that properly applied adversarially trained models can mimic 1-NN classifiers indicates that generalization powers of adversarially trained models might also mimic that of 1-NN classifiers. As 1-NN classifiers depend on distance metrics, test accuracy and test genuine adversarial accuracy of adversarially trained models can also be dependent on the distance metric used for measuring adversarial robustness. If robustness to perturbations on the distance measure is not much related to generalization, it is possible that there exists a tradeoff between test accuracy and test adversarial robustness.

We apply two analyses on MNIST (LeCun et al., 2010) and CIFAR-10 (Krizhevsky, 2009) data. In the first analysis, we analyzed different predictions on test samples between non-adversarially trained models and adversarially trained models to check if adversarially trained models resemble 1-NN classifiers. We measured significance[3] using two-sided McNemar's test (McNemar, 1947). We used publicly available models for comparisons. We used PGD-AT (adversarial training with projected gradient descent) (Madry et al., 2017) and TRADES (Zhang et al., 2019) models for adversarially trained models. In the second analysis, we checked the test accuracies of 1-NN classifiers based on training data to check globally robust models' generalization powers. $l_\infty$ norm-based 1-NN classifiers can take multiple nearest neighbors for each sample. In such cases, we calculated values ignoring these samples and using the class with the largest number of nearest neighbors for prediction. Values obtained by the latter approach are shown in parentheses in the tables. Details of the models and further results are in Section G in the appendix.

On MNIST (LeCun et al., 2010) data, PGD-AT ($l_\infty$ norm, $\epsilon = \frac{8}{255}$) model showed fewer agreements with 1-NN classifier than the non-adversarially trained model (shown in Table 1). This result conflicts with the speculation that adversarially trained models mimic 1-NN classifiers. However, considering the fact that no significant difference in proportions was found, this could be due to chance. (The fact that non-adversarially trained model already behave similarly with 1-NN classifier on test samples, shown in Table 5 in the appendix, could also be contributed to this result.) As shown in Table 2, on CIFAR-10 (Krizhevsky, 2009) data, adversarially trained models behaved more similarly to 1-NN classifier than non-adversarially trained models on test samples. The differences in the proportions were significant except for PGD-AT ($l_\infty$ norm, $\epsilon = \frac{8}{255}$). These results indicate that adversarial training can force models behave like 1-NN classifiers in some data. However, the observation that fewer than half of the changed predictions agree with 1-NN classifiers indicates

---

[1]"almost everywhere" is a term used in mathematical analysis. If a property holds almost everywhere, it means the subset of points that does not satisfy the property has measure zero.

[2]Properly applied adversarial training refers to adversarial training (Goodfellow et al., 2014) with no conflicting regions originating from overlapping regions. Adversarial training with Voronoi constraints (Khoury & Hadfield-Menell, 2019) satisfy this. Standard adversarial training with sufficiently small $\epsilon$ can satisfy this.

[3]Significance threshold 0.01 was used.

Table 1: Proportions of agreements with 1-NN classifiers among different predictions on test samples with non-adversarially trained models and adversarially trained models (Engstrom et al., 2019; Zhang et al., 2019) for MNIST (LeCun et al., 2010) training data.

| Model (Distance metric, training $\epsilon$) | PGD-AT ($l_\infty$ norm, $\epsilon = 0.3$) | TRADES ($l_\infty$ norm, $\epsilon = 0.3$) |
|---|---|---|
| Non-adversarially trained model | 0.1948 (0.3182) | 0.1296 (0.2407) |
| Adversarially trained model | 0.1039 (0.2597) | 0.2037 (0.2963) |
| p-value | 0.05408 (0.3966) | 0.4807 (0.7111) |

Table 2: Proportions of agreements with 1-NN classifiers among different predictions on test samples with non-adversarially trained models and adversarially trained models (Engstrom et al., 2019; Zhang et al., 2019) for CIFAR-10 (Krizhevsky, 2009) training data.

| Model (Distance metric, training $\epsilon$) | PGD-AT ($l_2$ norm, $\epsilon = 0.25$) | PGD-AT ($l_2$ norm, $\epsilon = 0.5$) | PGD-AT ($l_2$ norm, $\epsilon = 1.0$) | PGD-AT ($l_\infty$ norm, $\epsilon = 0.03137 = \frac{8}{255}$) | TRADES ($l_\infty$ norm, $\epsilon = 0.031$) |
|---|---|---|---|---|---|
| Non-adversarially trained model | 0.1529 | 0.1599 | 0.1996 | 0.1231 (0.1261) | 0.1064 (0.1103) |
| Adversarially trained model | 0.2562 | 0.2814 | 0.2948 | 0.1480 (0.1518) | 0.1795 (0.1878) |
| p-value | $1.593 \times 10^{-5}$ | $1.500 \times 10^{-8}$ | $6.403 \times 10^{-9}$ | 0.09110 (0.08525) | $8.202 \times 10^{-11}$ ($1.008 \times 10^{-11}$) |

that other factors are involved in the prediction changes. It is also possible that other changes in predictions occurred during the process of making the models behave like 1-NN classifiers as convolutional neural networks have limited capacities.

As shown in Table 3, 1-NN classifiers on MNIST data (LeCun et al., 2010) scored high accuracies. However, 1-NN classifiers scored low accuracies on CIFAR-10 data (Krizhevsky, 2009). Using different distance metrics resulted in different accuracies on both data. These results show that we might have to sacrifice generalization performance when seeking global robustness on training data depending on data and distance metrics.

When we combine the results, adversarial training can make models behave like 1-NN classifiers and make other changes. When 1-NN classifiers have low generalization performances, 1-NN resembling adversarially trained models can have reduced test accuracies. Other prediction changes can also contribute to reduced test accuracies.

## 5 RELATED WORKS

(Wang et al., 2018) showed that 1-NN classifier is non-robust when the exclusive class assumption (in the problem setting) is violated. However, they also showed that, 1-NN classifiers can be robust against adversarial attacks under the exclusive class assumption. Based on that result, they proposed a robust 1-NN classifier that uses a subset of the training data whose labels are highly confident.

Table 3: The values or ranges of test accuracies of 1-NN classifiers on training data for MNIST (LeCun et al., 2010) and CIFAR-10 (Krizhevsky, 2009) data.

| Data | MNIST | | CIFAR-10 | |
|---|---|---|---|---|
| Metric | $l_2$ norm | $l_\infty$ norm | $l_2$ norm | $l_\infty$ norm |
| 1-NN classifier | 0.9677 | [0.6142, 0.9662] ([0.8416, 0.8635]) | 0.3539 | [0.1722, 0.2427] ([0.1773, 0.2242]) |

(Khoury & Hadfield-Menell, 2019) showed that 1-NN classifiers can use exponentially fewer samples than standard adversarial training (Goodfellow et al., 2014) to correctly classifies samples on the tubular neighborhood. The authors also suggested an improved adversarial training method that uses Voronoi cell constrains instead of $\|\cdot\|$-ball constrains.

Under the settings that adversarial perturbations allowed to change classes, (Tsipras et al., 2018) and (Dohmatob, 2018) argued that the tradeoff between natural accuracy and standard adversarial accuracy may be inevitable in any classifier. Our work shows that this tradeoff can be avoidable by using genuine adversarial accuracy. Another explanation for the tradeoff is that robust classifications might require more model capacities than standard classifications (Nakkiran, 2019). Based on analysis on linear regression problems, (Raghunathan et al., 2019; 2020) suggested conditions of data distribution that adversarial training (Goodfellow et al., 2014) can hurt generalization even if only class consistent perturbations are used.

## 6 CONCLUSION

Even though standard adversarial accuracy is commonly used to measure the robustness of classifiers (Biggio et al., 2013; Madry et al., 2017; Tsipras et al., 2018; Zhang et al., 2019), it has problems that it allows overlapping regions. In this work, we introduce genuine adversarial accuracy that avoids such a problem. Hence, genuine adversarial accuracy does not have the tradeoff between accuracy and genuine adversarial accuracy unlike standard adversarial accuracy (Tsipras et al., 2018; Dohmatob, 2018; Zhang et al., 2019). Genuine adversarial accuracy ignores adversarial examples (Biggio et al., 2013) when applied perturbations are large enough to change classes of a classifier (1-NN classifier) as shown in Theorem 1. Hence, it does not favor a classifier with invariance-based adversarial examples (Jacobsen et al., 2019). These advantages of genuine adversarial accuracy suggest that graphs of genuine adversarial accuracies of classifiers need to be used for evaluating the robustness of classifiers. (In practice, it can be hard to calculate genuine adversarial accuracy by exact perturbation norm. However, one can use genuine adversarial accuracy by maximum perturbation norm explained in Section E of the appendix.)

Furthermore, in Theorem 2, we prove that a single nearest neighbor (1-NN) classifier is the most robust classifier according to genuine adversarial accuracy for given data and a distance metric by assuming exclusive class on data distribution. It suggests that adversarial training with Voronoi constraints (Khoury & Hadfield-Menell, 2019) would train hybrid models that seek optimal robustness on the training data. (For adversarial training with Voronoi constraints, one can also use soft labels explained in Section F and Section H of the appendix.)

As explained in Section 4, Theorem 2 gives us an insight that models trained by adversarial training (Goodfellow et al., 2014) without overlapping regions can mimic 1-NN classifiers. That was confirmed on analysis on CIFAR-10 data (Krizhevsky, 2009). That provides one possible factor why adversarial training might reduce test accuracy and why a tradeoff between test accuracy and test adversarial robustness might exist. As the choice of the distance metric can influence both the generalization power of 1-NN classifiers and the measure of adversarial robustness, one might have to be careful when choosing the distance metrics. In other to achieve both high test accuracy and test adversarial accuracy, distance metrics that are highly related to generalization might be necessary.

## 7 FUTURE WORK

Our analysis is based on the assumption that the data satisfy the exclusive class assumption in the problem setting. We also need similar analyses for various conditions that data violate that assumption. (Section H of the appendix shows a way of speculating optimally robust classifiers when data contain additive input noise.)

From Theorem 2, we know that there will be an almost everywhere unique optimally robust classifier (except for $\mathcal{X} - \mathcal{X}$ and Voronoi boundary $VB(\mathcal{X})$) for a given data and a distance metric. In addition, as explained in Section 4, using distance metrics that are highly related to generalization would be necessary to achieve both high test accuracy and test adversarial accuracy. Thus, we need to define a metric that captures various changes of images so that 1-NN classifiers based on that distance metric have high generalization power.

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
