# OpenReview forum: "Proper Measure for Adversarial Robustness"
_ICLR.cc/2021/Conference — Reject_

### Official Review · AnonReviewer3 · 2020-10-25
**Review for Proper Measure for Adversarial Robustness**

**Rating:** 3
**Confidence:** 4

**Review:**

This paper proposes a new measure called "genuine adversarial accuracy" for adversarial robustness of a classifier. The key idea is that 1) they calculate robust accuracy when the perturbation norm is exactly $\epsilon$, not smaller than $\epsilon$ like standard robust measures, and 2) each point will not be used multiple times in calculating the robust measure. Their theoretical results say that 2) is really the case, and an 1-NN classifier is most robust w.r.t. the proposed measure.

Strength of the paper
1. It provides a new robust measure, which potentially provides no trade-off solution for the adversarial robustness, if a distance leading to good generalization is properly chosen.
2. The information in Table 4 gives good insight of the benchimarking datasets, which can potentially benefit researchers of this literature.
3. Many interesting insights, such as regular adversarial training may push the model to behave similar to 1-NN classifiers on the training set, if $\epsilon$ is set small enough.

Some crucial technical flaws are out standing, and the paper is generally hard to read.

The key concern about the paper is the lack of clear definitions on the important concepts, and therefore, on the proposed robust measure. The authors may have a clear measure in their mind, but based on the provided definitions, it does not seem clear.

 1. The provided definition of Voronoi boundary $$VB(\mathcal{X})=\\{x'\in \mathbb{R}^d|\exists x_1,x_2\in \bar{\mathcal{X}}:x_1\neq x_2, \|x'-x_1\|=\|x'-x_2\|\\}$$ seems incorrect, or some examples on which their measure cannot be applied are not properly addressed. At least, I can find an example on which the proposed measure is not applied. For example, let's take a unit sphere as $\mathcal{X}.$ Consider $\ell_2$ norm. Then, for any combination of $x'\in \mathbb{R}^d$ and $x_1\in \bar{\mathcal{X}}$, we can find $x_2\in\bar{\mathcal{X}}$ s.t. $\|x'-x_1\|=\|x'-x_2\|$ and $x_2\neq x_1$ as long as $x'\neq x_1$. That means, for a unit sphere $\mathcal{X},$ the Voronoi boundary is the entire input space $\mathbb{R}^d$. Now, $VB(\mathcal{X})^c=\phi$, and therefore $S_{exact}(\epsilon)$ is not defined or empty in this case. Is this correct? Since having a sphere as a distribution support is not really pathological, e.g., C.I. for Gaussian distribution, I think this is a serious problem.

Because of this, \textbf{their key example in Figure 6, the illustration of genuine adversarial accuracy (GAA) seems to be incorrect. } I checked the proof for Figure 6 in Appendix A. In the provided calculation of the GAA in Figure 6, they say that the Voronoi boundary is $VB(\mathcal{X})=\{0\}.$ However, based on the their presented definition of Voronoi boundary, it seems to be incorrect. For example, let $x_1-=-1\in\bar{\mathcal{X}}.$ Then, for any point $x'\in(0,0.5),$ $\|x'-x_1\|=(x'+1)$. Define $x_2 =2x'+1$. Now we know that $x_2\in\bar{\mathcal{X}}$ and $\|x'-x_2\|=(x'+1)$. Therefore, $(0,0.5)$ must be a subset of $VB(\mathcal{X})$, and like wise $(-0.5,0)$ also must be. Therefore, at least $VB(\mathcal{X})\supset (-0.5,0.5).$ However, they caculate their results in Figure 6 by setting $VB(\mathcal{X})=\{0\}.$ Therefore, the presented entire results in Figure 6 seem to contain errors.

Likewise, for this reason, the theoretical results seem dubious.

2. Assuming somehow $VB(\mathcal{X})$ is well defined, e.g., by a re-definition, the set
$$S_{exact}(\epsilon) = \\{x\in\bar{\mathcal{X}}|\exists x'\in \mathcal{X}_{\epsilon}^c\cap VB(\mathcal{X})^c:\|x'-x\|=\epsilon\\} $$
over which the expectation is taken seems not representing the data very well. The perturbations set is not defined or empty for so many data points. To see this, bet an $\epsilon$-ball neighbor $B(\mathcal{X},\epsilon)=\\{x'|\|x-x'\|<\epsilon, x\in\bar{\mathcal{X}}\\}$

Now, your definition of $\mathcal{X}_{\epsilon}$ in page 5 is identical to

$\mathcal{X}_{\epsilon} = VB(\mathcal{X})^c\cap B(\mathcal{X},\epsilon)$,

which means $\mathcal{X}_{\epsilon}^c = VB(\mathcal{X})\cup B(\mathcal{X},\epsilon)^c$.

Therefore,
$$\mathcal{X}_{\epsilon}^c\cap  VB(\mathcal{X})^c = (VB(\mathcal{X})\cap VB(\mathcal{X})^c)\cup (B(\mathcal{X},\epsilon)^c\cap VB(\mathcal{X})^c) = B(\mathcal{X},\epsilon)^c\cap VB(\mathcal{X})^c\subset B(\mathcal{X},\epsilon)^c$$

Now, in the above definition of $S_{exact}(\epsilon)$, $x'$  must be reached by $x\in\bar{\mathcal{X}}$ by distance $\epsilon$. In other words, any element in $S_{exact}(\epsilon)$ must be able to reach to $B(\mathcal{X},\epsilon)^c$ by $\epsilon$ distance. For example, if $\mathcal{X}$ is again a unit sphere, then $S_{exact(\epsilon)}$ is a subset of the surface of the sphere. Now, no matter how the classes are distributed \emph{inside} the unit sphere, the genuine robust measure only consider the \emph{surface} of the sphere to measure the robustness.


The second major concern is that the experiment section (section 4) seems not ready yet. It contains unsupported/self-contradicting arguments. Also, some additional information about the experiment conducted seems necessary.
1. \textit{This result gives us an insight that properly applied adversarial training(adversarial training (Goodfellow et al., 2014) with no conflicting regions originating from overlapping regions) will train models to mimic 1-NN classifiers as their training loss try to enforce them to make the same prediction as 1-NN classifiers}: The authors argue that this training's loss forces to predict on the training set as like 1-NN classifiers, which is optimal w.r.t. the genuine robust accuracy. The author may need to provide that when $\epsilon$ is small, minimizing the standard adversarial loss is identical to maximize genuine robust accuracy. Otherwise, this argument does not make sense.
2.  \textit{If robustness to perturbations on the distance measure is not much related to generalization, it is possible that there exists a tradeoff between test accuracy and test adversarial robustness}: Why is that? This argument is confusing because so far the generalization is w.r.t. robustness, but all of sudden the authors are talking about tradeoff between natural accuracy and robustness. So far, you said that for a good distance metric, 1-NN classifiers will be robust also on the test dataset. Now, what are the missed sentences to get your conclusion on the trade-off?
3. How do you define and measure proportions of agreements with 1-NN classifiers for Table 1 and 2? Do you test a point from the training set after adding some perturbations or not? If you add some perturbations, then how do you generate them?
4. The result in Table 2 (among different predictions) is very different from Table 6 (on whole predictions). What is the definition of and difference between "among different predictions" and "on whole predictions"? Why does one show significantly different agreements, whereas the other show no almost no difference?
5. Along the same line, the numbers in Table 2 are way larger than the numbers in Table 5. Why is that?
6. In Table 1 and 2, I see that the agreement proportion values for non-adversarially trained models are different. For example, in table 2, 0.1529 for PGD-AT $\epsilon=0.25$ and 0.1996 for PGD-AT $\epsilon=1.0$. I guess that for PGD-AT $\epsilon=0.25$ and $\epsilon=1.0$, the non-adversarially trained model would be $\epsilon=0$ for both models. Why do we see this difference? If they were not adversarially trained, and if they all have the same model architecture, these observed differences may challenge to the solidity of the experiment part.

Other concerns:
1. The main contribution of this paper is unclear. First, the usefulness of this measure is unclear. We can not use this measure to measure the robustness of models as the authors do not provide a way to measure this genuine adversarial accuracy. They only compare the prediction of classifiers to that of 1-NN classifiers, which is by Theorem 2 optimal on the training set. Then, if it is difficult to measure genuine robust accuracy, then do you think the major usage of your measure is to show the lack of model capacity used for robust learning? Or, what is a main benefit of this proposed measure even in the absence of the specific algorithm to actually measure it?
2. I do not see the theory and experimental results are coherent to their final conclusion. If I understand correctly, originally, the first experiment was intended to show the high agreement between 1-NN classifiers and adversarially robust models. By the argument of the authors, since the $\epsilon$'s used for PGD-AT or TRADES are small enough based on Table 4, these trainings can be seen properly applied adversarial training. Then, the adversarially trained models on the training set should behave similarly to 1-NN classifiers. However, not as expected, in Table 1,2,5, and 6 the portion of agreement is at best about 0.6 for MNIST and 0.36 for CIFAR-10. Since those numbers are not close to 1, i.e., lower than as expected by Theorem 2, they are speculating these low numbers are due to the limited network capacities. However, in Conclusion, they say that \textit{" can mimic 1-NN classifiers. That was confirmed on analysis on CIFAR-10 data"}, which is perplexing.

Suggestions and Questions:
1. The definition of $VB(\mathcal{X})$ could have been refined and some intuition about this definition could have been provided. Also, could you give a justification of your robust measure in my unit sphere example in 2?
2. W.r.t. Table 4, I think it will be even more beneficial if the authors can provide the maximum of the minimum distance within the same class, across or within the training and test set. It will help to understand the robustness of 1-NN classifiers and the performance gab between 1-NN classifiers and other network based classifiers, either robust or not.
3. The main body can be revised in a way that the major contribution is clear and extra information is located in a proper location. For example, too many new/necessary information is at the conclusion even in a parenthesis.
4. In conclusion, for the sentence \textit{"That provides one possible factor ...why a tradeoff between test accuracy and test adversarial robustness might exist.", }can you explain more about it? In your paper, I personally did not get any possible factor why a tradeoff happens.

---

> ### Author Response · Authors · 2020-11-16
> **Answers to the theoretical concerns**
>
> We appreciate the reviewer for detailed and throughout review. We apologize for the errors and insufficient explanations.
>
> First, we answer the theoretical concerns. As the reviewer used the expression “the surface of the sphere”, we guess the reviewer meant a filled sphere (or a unit ball) when sphere is mentioned. We answer the theoretical concerns based on that. Without the loss of generality, we assume the central point of the sphere is located at zero.
>
> 1. The definition of Voronoi boundary was incorrect in the current version. As you pointed out, the definition is problematic when we consider situations that data have uncountably many points.
> We need to define (open) Voronoi cell of a sample $x\in\bar{\mathcal{X}}$ as $Vor(x)=\left\lbrace x'\in\mathbb{R}^d|\ \left\||x-x'\right\||<\left\||x_{clean}-x'\right\||,\forall x_{clean}\in \bar{\mathcal{X}}-\left\lbrace x\right\rbrace\right\rbrace$. Then, the corrected definition of Voronoi boundary $VB(\mathcal{X})$ is $\left(\bigcup\limits_{x\in\bar{\mathcal{X}}}{Vor(x)}\right)^{\mathsf{c}}=\bigcap\limits_{x\in\bar{\mathcal{X}}}{ Vor(x)^{\mathsf{c}} }$.
>
> Based on the redefinition, in the unit sphere example, $Vor(x)= \left\lbrace kx|k\ge1\right\rbrace$ for $\left\||x\right\||=1 $, and $Vor(x)=\left\lbrace x\right\rbrace$ for $\left\||x\right\||<1$. Then, Voronoi boundary is $\varnothing$.
>
> In the one-dimensional example, $Vor(x)= \left\lbrace x \right\rbrace$ for $x\in\left(-2,-1\right)\cup {\left(1,2\right)}$, $Vor(-2)=\left(-\infty,-2\right]$, $Vor(-1)=\left[-1,0\right)$, $Vor(1)=\left(0,1\right]$, and $Vor(2)=\left[2,\infty\right)$. Voronoi boundary $VB(\mathcal{X})=\left\lbrace 0\right\rbrace$.
>
> 2. The situation that $S_{exact}(\epsilon)$ is quite small compared to $\bar{\mathcal{X}}$ was due to the property that genuine robust measure avoids already calculated points for smaller epsilon.
> Standard adversarial accuracy was devised to only consider a single epsilon $\epsilon$. Genuine adversarial accuracy was devised to consider all epsilon value (to handle problems of using a single epsilon as explained in the sunset dataset). Hence, we need to consider other $\epsilon$ in genuine adversarial accuracy to see the whole picture of the robustness.
>
> When $\epsilon=0$, all point inside the filled sphere is used for calculating genuine adversarial accuracy, which is just the natural accuracy for $\epsilon=0$. When $\epsilon>0$, the perturbed samples $x’$ starting from the points $x$ inside the filled sphere either located in the other points in $\mathcal{X}$ or $\mathcal{X}^{\mathsf{c}}$.
> When $x’\in\mathcal{X}$, (i) $c_{x'}=c_x$ or (ii) $c_{x'}\neq c_x$. In the former case (i), using $x'$ for calculating adversarial accuracy gives the same (redundant) information as in $\epsilon=0$. In the latter case (ii), using $x'$ for calculating adversarial accuracy gives the conflicting information from $\epsilon=0$, and thus, if we use these points in adversarial accuracy for $\epsilon>0$, then such adversarial accuracy will favor models with invariance-based adversarial examples (as models need to output the original class $c_{x}$ even if the class $c_{x'}$ was changed).
> When $x’\in\mathcal{X}^{\mathsf{c}}$, there exists a clean sample $x_{clean}=\frac{x'}{||x'||}$ in the sphere surface such that $||x'-x_{clean}||<||x'-x||=\epsilon$. Using $x'$ for calculating adversarial accuracy gives either redundant information (when (iii) $c_{x}=c_{x_{clean}}$) or conflicting information (when (iv) $c_{x}\neq c_{x_{clean}}$). To avoid using redundant or conflicting information, we ignore such $x$ and only consider the point on the surface ($x_{clean}$ in this paragraph) in the definition of $S_{exact}(||x'-x_{clean}||)$. (We used $x_{clean}$ rather than $x$ in this paragraph because $x_{clean}$ is closer to $x'$.)
>
> If we give a more extreme situation, you can consider when $\mathcal{X}=\mathbb{R}^d$. In that case, $S_{exact} (\epsilon)=\varnothing$ when $\epsilon>0$. In this case, the oracle classifier just needs to accurately classify samples on $\mathcal{X}$ as $\mathbb{R}^d=\mathcal{X}$. Every perturbation (with $\epsilon>0$) gives redundant or conflicting information from $\epsilon=0$. (To avoid getting redundant or conflicting information, $S_{exact} (\epsilon)$ should be $\varnothing$ when $\epsilon>0$.)

---

> ### Author Response · Authors · 2020-11-16
> **Answer to the questions about the experiment section (section 4)**
>
> We answer questions regarding the experiment section (section 4). In most cases, when similarity with the 1-NN classifier and generalization were mentioned, we meant relative similarity and relative generalization of robust classifiers compared to standard classifiers. We didn’t intend to explain the absolute similarity or absolute generalization of adversarially trained models.
> 1. We will explain or proof that in the later versions.
> 2. Poor distance metric means robustness to perturbations on the distance metric is not much related to generalization. In other words, using poor distance metrics means the 1-NN classifier based on the distance metric generalizes poorly (as 1-NN classifier is optimally robust for the given distance).
> The test adversarial robustness requires robustness. When we use a poor distance metric, robust classifiers are (locally or globally) similar with the 1-NN classifier, and thus they generalize relatively poorly (have relatively less test accuracy). When poor distance metric was used, to increase the test adversarial robustness, there might be decrease in test accuracy and we have the trade-off (in testing time).
>
> We experimentally showed poor generalization powers of the 1-NN classifiers based on $l_2$ and $l_{\infty}$ norm distances (in Table 3). We experimentally showed the relatively more agreements (with 1-NN classifiers) of adversarially trained models in CIFAR-10 data (in Table 2). We used the term 'one factor' as only $0.1480$ to $0.2948$ of adversarially trained models' predictions (among different predictions) agree with the 1-NN classifiers. The process of increasing agreements could lead to the change of predictions and the relatively increased test errors of the adversarially trained models.
>
> 3. For Table 1 and 2, we first found different predictions on (clean) test samples between non-adversarially trained models and adversarially trained models. The proportions of agreements (with 1-NN classifier) is calculated by $\frac{\text{Agreements of the model with 1-NN classifier (among different predictions)}}{\text{Total number of different predictions}}$. We tested only clean test samples as we wanted to show the agreements (with 1-NN classifier) on the data distribution.
>
> 4, 5, 6. When the proportions of agreements were calculated, "among different predictions" meant that we calculate using the formulation shown above. In this case, denominator was the number of different predictions between non-adversarially trained models and adversarially trained models. (Answer for 6:) Even if we used the same non-adversarially trained model for PGD-AT $\epsilon=0.25$ and PGD-AT $\epsilon=1.0$, in Table 2, we used different adversarially trained models. The denominators were different, and thus, we got different values $0.1529$ and $0.1996$. (Numerators might also different, but we guess difference in denominator was sufficient to explain the difference in the proportion values.)
> On the other hand, "on whole predictions" meant we calculate the proportions by $\frac{\text{Agreements of the model with 1-NN classifier}}{\text{Number of test samples (that 1-NN classifier’s prediction is defined)}}$. In this case, denominators were the number of test samples ($10000$ for $l_2$ norm, and vary for $l_{\infty}$ norm due to tied predictions).
> The more difference of agreements in Table 2 than Table 6 (and also Table 1 than Table 5) was due to relatively small denominators for Table 2 (and Table 1).
>
> Some of the values in Table 1 and 2 can be derived from Table 7 to 13. For instance, PGD-AT $l_2$ norm $\epsilon=0.25$ in Table 2, $0.1529$ is calculated by $\frac{111}{726}$. ($111$ and $726$ are from Table 9.) PGD-AT $l_2$ norm $\epsilon=0.25$ in Table 2, $0.2562$ is calculated by $\frac{186}{726}$. ($186$ and $726$ are from Table 9.)
>
> To give an example for the calculations, let us consider a situation when predictions on test samples are $[0,0,2,0,0,1]$, $[0,0,1,1,0,2]$ and $[0,0,1,1,2,2]$ for non-adversarially trained model, adversarially trained model, 1-NN classifier, respectively. ($\mathcal{Y}=\lbrace 0,1,2\rbrace$.) Then, the proportions of agreements (with 1-NN classifier) among different predictions are $\frac{0}{3}$ for non-adversarially trained model and $\frac{3}{3}$ for adversarially trained model. The proportions of agreements (with 1-NN classifier) on whole predictions are $\frac{2}{6}$ for non-adversarially trained model and $\frac{5}{6}$ for adversarially trained model. The difference in "among different predictions" case is larger in this example ($\frac{3}{3}-\frac{0}{3}=1>0.5=\frac{5}{6}-\frac{2}{6}$).

---

### Official Review · AnonReviewer2 · 2020-10-27
**Unclear contribution**

**Rating:** 3
**Confidence:** 5

**Review:**

The authors study the question of adversarially robust classification---classifying inputs under worst-case norm-bounded corruptions. The authors argue that the standard notion of robust accuracy is inadequate for properly evaluating model performance and propose an alternative. Then, they show that for any fixed set of data points, the 1-Nearest-Neighbor classifier is the optimally robust classifier according to their proposed measure. Finally they measure empirically how similar existing robust classifiers are to the 1NN classifier.

While the paper studies an interesting problem, the impact of its contributions is unclear:
- **Inadequacy of robust accuracy.** The authors provide a number of toy examples that are unfortunately not convincing. In these examples, the notion of ground-truth label is undefined on most of the input space and the authors resort to arguments about an "oracle classified" and "invariance-based adversarial examples" which are not rigorously defined here. For instance, in the one dimensional examples, why is f2(x) not a good solution? It is perfectly accurate and robust up to eps=1. Similarly, in the sunset example, why is the top point classified as blue? Based on my understanding, in this dataset, the maximally robust classifier is exactly the max margin classifier and there is no trade-off between robustness and accuracy.
- **Alternative notion of robust accuracy.** I was unable to understand the proposed definition at an intuitive level. How does it defer from standard accuracy? Why is it better? What does it mean to "use a point more than once when calculating robustness"?
- **The 1NN classifier is maximally robust.** Since this statement if proved for a fixed set of datapoints, I do not understand its significance. Clearly, if we are given a set of points with their labels and we are asked to predict these labels *on the exact same set of points*, then using a 1NN classifier makes sense. But clearly, for many realistic machine learning problems (e.g., images) NN classifiers will inevitably perform poorly due to their inability to account for natural data invariances (e.g., image crops).
- **Similarity of adv. trained models with the 1NN classifier.** I do not see the significance of this experiment. For instance, on CIFAR10, the 1NN classifier performs quite poorly so I'm not sure what this comparison is meant to convey.

Overall, the paper need to significantly improve in focus, stating concrete and relevant research questions that its contributions are meant to answer.

Other comments (not affecting score):
- [This paper by Suggala et al.](https://arxiv.org/abs/1806.02924) also explores alternative notions of robust accuracy and are quite relevant.
- There are several grammar issues  throughout the manuscript.

---

### Official Review · AnonReviewer1 · 2020-10-28
**Lack of significant results**

**Rating:** 3
**Confidence:** 4

**Review:**

The paper introduces a novel metric for measuring adversarial accuracy of machine learning models, dubbed genuine adversarial accuracy. This is motivated by standard adversarial accuracy favouring in some instances classifiers with higher vulnerability to invariance-based adversarial examples.

One of my main reservation about this paper is the lack of significance. The genuine adversarial accuracy metric seems to be motivated by corner cases; the paper does not demonstrate the importance of this metric neither for practical evaluations of adversarial robustness nor for significant theoretical purposes.

Further observations:
- In Definition 2: I don’t think it is common to define adversarial accuracy w.r.t. a specific loss function.
- References are repeated numerous times, e.g. any mentioning of “adversarial training” is followed by the “(Goodfellow 2014)” reference, which is redundant and affects readability.
- In 1.2.1: I don’t see why the oracle classifier will be f_3 or why it should even be uniquely determined.
- In 1.3: The objective of adversarial training is not necessarily to improve accuracy on clean samples.
- In 1.3.1: “is consists of” -> “consists of”
- In Definition 3: What does it mean to “use exact perturbation norm”? The type setting could be improved by not putting equations under bullet points.
- Avoiding “conflicts” of using points twice may not be required if those points have measure zero.
- Theorem 1 immediately follows by definition - this is more of a proposition the proof of which is obvious.
- Also Theorem 2 is obvious, and Section 3 consists of only one paragraph.
- Section 4 is pretty vague and inconclusive. The argumentation relating adversarial training to generalisation of 1-KNN classifiers isn’t convincing.
- The conclusions make several references to different parts of the Appendix which haven’t been discussed in the main body of the paper.

---

### Official Review · AnonReviewer4 · 2020-10-29
**The paper aims to a new definition of adversarial correctness, but the benefits (and even the exact definition itself) are not clear**

**Rating:** 3
**Confidence:** 5

**Review:**

Summary:

The paper revisits previously known definitions of "adversarial accuracy" (and its complement: adversarial risk) which captures the accuracy (and risk) of learning models under adversarial perturbations of the test instances. The paper argues that the established (called standard) definition of adversarial accuracy has issues that need to be resolved, and then this work proposes a new definition (called genuine adversarial accuracy). The main issue they mention with previous definition is that they come at odds with accuracy. Their definition aims to fix this by giving an alternative way of defining accuracy. They then study the robustness of 1-NN under the newly proposed definition and claim that this algorithm is the most robust classifier according to this new definition. Proofs are differed to supplemental material.


######################################################

Summary of reasons for the score:

Unfortunately, the paper’s treatment of the subject is not formal enough. The new definition also does not seem to be an improvement in any clear way. There is no given clear intuition for the new definition (which is quite complex to state) and what is exactly achieved. The theorem statements that are based on the definition are not formal either.


##########################################################

Pros:

Understanding the adversarial robustness (e.g., generalization of adversarially trained models) from a theoretic perspective is a very important and nontrivial problem. So, alternative (new) definitions that might allow us to understand the picture better are potentially highly valuable. This paper aims to improve the state of the art along this direction.

#############################################################

Cons:


1.	Unfortunately, the paper does not clearly set objective goals to be achieved. The "issues" with previous definitions are not clearly discussed.

2.	There are issues with the level of formality of the stated results, and the proposed definition is too complex to even read. No clear evidence is presented to justify the usefulness of the new definition in resolving challenges that previous definitions do not. A work aiming to settle definitional issues needs to be a lot more formal and precise.

3.	Previous work has addressed some of the issues that the paper seems to be aiming to address. The papers [1,2] below (not cited) already point out a simple variation of the definition of adversarial risk under which there is no trade-off between accuracy and robustness. A simple way to prevent the definition of robustness to be at odds with accuracy is to require the adversarial sample (x^*) to be a misclassification. Also note that all these variants of definitions are equivalent to the main-stream definition (Def 1 in the paper), if one assumes that the underlying concept function (i.e., the ground truth) remains robust under the allowed amount of perturbation. So, in contexts such as image classification in which human is the ground truth and is already assumed to be robust to the amount of allowed noise/perturbation, the mainstream definition that is used has no issues. Issues arise when such definition are applied to a *different* context in which the ground truth might *not* be robust under the amount of allowed perturbation.

[1] Suggala et al. "Revisiting adversarial risk." aistats 2019.

[2] Diochnos et al. "Adversarial risk and robustness: General definitions and implications for the uniform distribution." NeurIPS 2018.

#######################################################

Main comments and questions:

The abstract states that:
"Based on this result, we suggest that using poor distance metrics might be one factor for the tradeoff between test accuracy and l_p norm-based test adversarial robustness."
I am not sure how this conclusion was obtained. On a related note, I do not follow how Section 4 is expanding on this thesis.

Note that l_p norm-based attacks are used since bounded l_p norms of perturbation does not seem to change human's judgement (there might be other perturbations that have this property too) so an *attack* under such limited perturbations is already an issue that needs to be addressed (but using such metrics for positive results could be questioned).

In problem setting page 1: why do you pick and fix the cross-entropy loss? It seems you want a general treatment of the definition of adversarial examples, and so other losses could be used as well?

I think definition 2 is confusing two different notions: adversarial loss and surrogate losses used for training for computational reasons.
Once a candidate classifier f is trained, to know its risk/error under adversarial attacks, one shall *not* pick x* based on a loss such as cross-entropy loss. x* is simply a point in the Ball of allowed perturbations around x such that f(x*) is different from c(x) (assuming that c(x)=c(x*) – this means x* is also misclassified). A different question is how to train a model with minimum empirical adversarial loss. To do so, one needs to use computationally efficient approaches that might lead to using some surrogate loss instead of the 0/1 loss.

top of page 2:
"f2 and f3 are equally robust according to this measure. Thus, the maximum norm-based standard adversarial accuracy function cannot tell which classifier is better."
I cannot follow the argument here.

In Def 3:
Consider a case where X contains a ball of non-zero volume. Then, *every* point in R^d will belong to VB(X), as for every x' there will be two different points x1, x2 in X that have the same distance to x'. Then VB(X)^c will be empty, which makes your X_\eps (allowed perturbation sets) to be empty. Is that so?

In your notation in the first bullet of page 5, what is the distribution over S_{exact(\eps))? Note that stating the set alone is not enough to figure out what the expected value means, and since we are dealing with continuous settings the exact distribution is important to be state.

Theorem 1 is not formally stated. "adversarial accuracy only use each point only once" is not a mathematically well-defined statement (especially in the continuous-domain regime). If you desire concrete properties that your definition implies, state those properties mathematically (e.g., if you are talking about a probability here, write it down formally).

Theorem 2 is not formally stated. It is not clear what "it is the almost everywhere unique1 classifier that satisfies" means. What is the probability measure space here? You seem to say that "with measure one" in some probability space "(1-NN) classifier maximizes genuine adversarial accuracy..." if this is correct, the theorem needs to be restated differently.

#########################################################################

More minor comments and typos:

The notation a_{std; max}(\epsilon) is missing (and shall somehow denote) the function (f1 or f2) as well, as it depends on the function.

page 3
"adversarial training try to" -> "tries to"

pag3 4:
"dataset is consists" -> remove "is"
"The other class (class B) is consist of" -> consists of

page 4: the points on the red circle are denoted as red, but you said the points "points inside the circle classified as class A". I guess you mean the points on and inside the circle are in class A?


>> commented after the rebuttals:
Thanks for the answers. However, I think the paper still suffers from too many basic issues (still standing from the review). Main one is that it is not clear what the current definition is and what it achieves that previous definitions miss. There are informal texts, but they are not coherent formal and verifiable.

---

### Decision · Program_Chairs · 2021-01-07
**Final Decision**

**Decision:**

Reject

**Comment:**

The paper considers new notions of adversarial accuracy and risk which are called "genuine" with an aim to fix issues with the existing definitions in the literature. A number of issues in the paper, including lack of motivation and intuition, and poor formalism were identified by the reviewers. The paper also fails to cite some of the previous literature that has identified similar issues. The authors have only responded to some of the questions raised by the reviewers.